# The role of gadolinium in magnetic resonance imaging for early prostate cancer diagnosis: A diagnostic accuracy study

Ilinca Cosma[1,2], Cornelia Tennstedt-Schenk[3], Sven Winzler[2], Marios Nikos Psychogios[4], Alexander Pfeil[5], Ulf Teichgraeber[1], Ansgar Malich[2], Ismini Papageorgiou[1,2]*

1 Institute of Diagnostic and Interventional Radiology, University Hospital Jena, Jena, Germany, 2 Institute of Radiology, Suedharz Hospital Nordhausen, Nordhausen, Germany, 3 Institute for Pathology, Muehlhausen, Germany, 4 Department of Neuroradiology, Clinic of Radiology and Nuclear Medicine, University Hospital Basel, Basel, Switzerland, 5 Department of Internal Medicine, University Hospital Jena, Jena, Germany

* ismini.e.papageorgiou@gmail.com

## Abstract

### Objective

Prostate lesions detected with multiparametric magnetic resonance imaging (mpMRI) are classified for their malignant potential according to the Prostate Imaging-Reporting And Data System (PI-RADS™2). In this study, we evaluate the diagnostic accuracy of the mpMRI with and without gadolinium, with emphasis on the added diagnostic value of the dynamic contrast enhancement (DCE).

### Materials and methods

The study was retrospective for 286 prostate lesions / 213 eligible patients, n = 116/170, and 49/59% malignant for the peripheral (Pz) and transitional zone (Tz), respectively. A stereotactic MRI-guided prostate biopsy served as the histological ground truth. All patients received a mpMRI with DCE. The influence of DCE in the prediction of malignancy was analyzed by blinded assessment of the imaging protocol without DCE and the DCE separately.

### Results

Significant (CSPca) and insignificant (IPca) prostate cancers were evaluated separately to enhance the potential effects of the DCE in the detection of CSPca. The Receiver Operating Characteristics Area Under Curve (ROC-AUC), sensitivity (Se) and specificity (Spe) of PIR-ADS-without-DCE in the Pz was 0.70/0.47/0.86 for all cancers (IPca and CSPca merged) and 0.73/0.54/0.82 for CSPca. PIRADS-with-DCE for the same patients showed ROC-AUC/Se/Spe of 0.70/0.49/0.86 for all Pz cancers and 0.69/0.54/0.81 for CSPca in the Pz, respectively, *p*>0.05 chi-squared test. Similar results for the Tz, AUC/Se/Spe for PIRADS-without-DCE was 0.75/0.61/0.79 all cancers and 0.67/0.54/0.71 for CSPca, not influenced by DCE (0.66/0.47/0.81 for all Tz cancers and 0.61/0.39/0.75 for CSPca in Tz). The added Se and Spe of DCE for the detection of CSPca was 88/34% and 78/33% in the Pz and Tz, respectively.

**Data Availability Statement:** All relevant data are within the manuscript and its Supporting Information files.

**Funding:** The author(s) received no specific funding for this work.

**Competing interests:** The authors have declared that no competing interests exist.

**Abbreviations:** ADC, Apparent Diffusion Coefficient; ASAP, Atypical Small Acinar Proliferation; AUC, Area Under Curve; CSPca, Clinically Significant Prostate cancer; DCE, Dynamic Contrast Enhancement; DWI, Diffusion-Weighted Imaging; IPca, Insignificant Prostate cancer; MRI, Magnetic Resonance Imaging; mpMRI, multiparametric MRI; PI-RADS™, Prostate Imaging Reporting and Data System; Pz, Peripheral zone; ROC, Receiver Operating Characteristic; T1w, T1-weighted imaging; T2w, T2-weighted imaging; TRUS, Transrectal Ultrasound-guided; Tz, Transitional zone.

## Conclusion

DCE showed no significant added diagnostic value and lower specificity for the prediction of CSPca compared to the non-enhanced sequences. Our results support that gadolinium might be omitted without mitigating the diagnostic accuracy of the mpMRI for prostate cancer.

## Introduction

Magnetic resonance imaging (MRI) is a non-invasive and accurate diagnostic method for the early diagnosis of prostate cancer (Pca) [1]. Especially in the last 3 years, MRI opts to replace the Transrectal Ultrasound-guided (TRUS) biopsy and become the standard of care for the early diagnosis of Pca in patients with elevated Prostate-Specific Antigen, while still maintaining its role in follow up, active surveillance and staging. The European Association of Urology, European Society for Radiotherapy and Oncology and International Society of Geriatric Oncology guidelines propose the role of MRI in early prostate cancer diagnosis in view of an MR-guided biopsy in clinical scenarios with persistent suspicion for malignancy after at least one negative TRUS biopsy [2]. This approach is the most acceptable, supported by the European Society of Medical Oncology and the British National Institute for Health and Care Excellence as well [3,4]. In order to normalize the image interpretation language to a common denominator, the European Society of Urogenital Radiology and the American College of Radiology released a structured reporting system, the Prostate Imaging Reporting And Data System [5] updated to PI-RADS™2 in 2015 and PI-RADS™2.1 in 2019 [1,6–8]. PI-RADS requires a diagnostic standard of anatomical (T2-weighted, T2w) and functional sequences (Diffusion-Weighted Imaging, DWI), including a series of Dynamic Contrast Enhancement (DCE). The combined protocol (T2w, DWI, and DCE) is summarized as a multiparametric MRI (mpMRI).

Although the utility and diagnostic value of contrast enhancement was enthusiastically endorsed in the first steps of structured prostate imaging [5,9,10], increasing demand in prostate MRI examinations [11], debated issues such as the gadolinium toxicity and tissue deposition [12,13] as well as the cost inflation with scanning time and use of gadolinium have stimulated the community to re-assess the added value of the DCE. Independent research groups converge towards the opinion that DCE has no significant added value in diagnostic accuracy [14–23]. However, the field remains heavily debated by datasets that support the DCE value in the diagnosis of clinically significant Pca (CSPca) towards the insignificant Pca (IPca) [24–28], especially in the hands of inexperienced readers [28] or for smaller lesions [29]. Currently, DCE is a standard recommendation in the most recent update of the prostate imaging guidelines (PI-RADSv2.1) [7,8] and a common practice for many radiological units.

The current study aims to assess the DCE necessity in the mpMRI protocol for the first Pca diagnosis using a retrospective database. The independent and added value of DCE was evaluated for all cancers and CSPca separately in the peripheral (Pz) and the transitional (Tz) prostate zones. Overall, we provide evidence that the added value of the DCE is not statistically significant, and gadolinium could be omitted without hampering the diagnostic accuracy of mpMRI.

## Materials and methods

### Ethical statement

Data were analyzed retrospectively, fully anonymized, following the ethical standards laid down in the 1964 Declaration of Helsinki and its amendments as well as the European

Regulation 536/2014. The Institutional Review Board of the University Hospital of Jena approved the study and waived the mandate from obtaining a legally valid informed consent from the included subjects (6/2019) [30].

## Study design and participant flow

The study was designed according to the Standards for Reporting of Diagnostic Accuracy (STARD) guidelines [31] and included n = 286 lesions from N = 213 eligible patients aged 64 ±7 years (mean/$\sigma$), screened with mpMRI in our department between 1/2012 and 11/2017 (S1 Fig). The MRI was conducted upon clinical suspicion for prostate cancer based on an elevated PSA assay and, in the vast majority, after an inconclusive transrectal ultrasound-guided biopsy. The diagnostic MRI was conducted at least four weeks after the ultrasound-guided biopsy to avoid artifacts. An MR-guided biopsy followed within three months post-diagnosis (mean/$\sigma$ = 40/38 days) and served as the histological ground truth. After the MRI-guided prostate biopsy, no further ultrasound-guided biopsies followed. In N = 6 patients with negative first MRI-guided biopsy and persisting clinical Pca suspicion, the MRI-guided biopsy was repeated within a time interval in 612±231 days (mean, standard deviation). From a total of 225 patients, we excluded 12 patients due to a lack of mpMRI before MRI-guided prostate biopsy (total eligible patient/lesions N/n = 213/286). One hundred seventy lesions (59%) derived from the transitional zone (74 malignant and 41 CSPca) and 116 lesions (41%) from the peripheral zone (79 malignant and 48 CSPca). The flow of participants in the study is thoroughly described according to the STARD guidelines in the supplement (S1 Table).

## Imaging protocol

Ten patients (15 lesions) were examined in a 1.5T MRI-system and the rest in a 3.0T-setup using a superficial multi-array coil (Philips Ingenia, Philips Medical Systems, Böblingen, Germany). The following protocol was applied as the standard of diagnosis with an average duration of 20 min without DCE and 35–40 min with DCE (S2 Table): a T2w turbo spin-echo (T2wTSE HR) in 2mm resolution, a DWI at 5 different b-values (b0-100-500-800-1000 s/ mm$^2$) and a T1-weighted (T1w) Fast Field Echo with DCE in 25 repetitions with 13.35 s temporal resolution and 7 s delay [29]. A weight-adjusted bolus of gadoteridol (ProHance®, Bracco Imaging S.p.A., Konstanz, Germany) 0.1mmol/kg was injected at 3 ml/min flow rate.

## Image evaluation

Two radiologists, one with intermediate experience (IP, 5th year of training in radiology) and a board-certified radiologist (AM with more than 15 years of board certification for prostate MRI), graded all lesions according to the qualitative criteria of PI-RADS™2 for T2w, DWI, and DCE [1,30]. The grading of non-enhanced sequences (T2w, DWI) on the basis of the 5-point Likert scale, which stratifies the level of suspicion for malignity (Supplemental information spreadsheet), was performed without the influence of DCE.

The DCE was graded separately by the same radiologists in a blinded manner to the T2w and DWI sequences [1] within a time interval of a minimum of one week. DCE scoring was binomial, based on the PIRADSv2 criteria, described as "positive" or "negative"depending on the speed and amplitude of the wash-in phase using the software DynaCAD v2 (Invivo, Gainesville, FL, USA). The DCE does not influence the final score of PI-RADS 4 or 5 lesions according to the PI-RADS™v2 criteria and is only relevant for triaging ambiguous lesions (PIRADS 3) in the Tz. However, we extended our analysis to include PI-RADS 4 and 5 lesions to assess the putative role of DCE in detecting CSPca in candidates for MR-guided prostate biopsy.

The retrospective analysis was based on the joint opinion because the separate reports were not accessible.

## MR-guided prostate biopsy

All patients were scheduled for an MRI-guided prostate biopsy in 40±38 days (mean/$\sigma$) from the initial assessment. A prophylactic antibiotic schema with fluoroquinolones starting 24 hours before the biopsy and a coagulation screening (international normalized ratio, partial thromboplastin time, and platelet count) were applied as a standard of care before the biopsy. The biopsy was performed at the same field strength with the diagnostic imaging, using the compatible, minimally invasive biopsy device DynaTRIM and its dedicated software Dyna-CAD (Invivo, a Philips Healthcare Company, Best, The Netherlands) to obtain an average of 2 biopsies per lesion. The size of biopsied lesions varied between 5 and 57 mm (S1 Table).

## Statistics and data analysis

Logistics and descriptive statistics were performed with the Microsoft Office suite 365 (Microsoft Ireland Operations Limited, Dublin, Ireland). The receiver operating characteristics (ROC), Analysis of Variance (ANOVA) with Dunn´s post-hoc test and Mann Whitney rank-sum test were performed with the Statistical Package for the Social Sciences version 25 (IBM GmbH, Ehningen, Germany). The Shapiro-Wilk test was used for validating the normal distribution hypothesis. The threshold for statistical significance was set at 0.05 ($\alpha = 0.05$). Outliers were included in the data analysis and not treated separately. Percentages are rounded up to the closest integer only for reporting purposes. Graphical processing of vectorized images and halftones was accomplished using Inkscape (GPL v2+, https://inkscape.org).

# Results

## Dynamic contrast enhancement has a low sensitivity for prostate cancer detection

T2w and DWI are the leading sequences for Pca diagnosis in the Tz and Pz, respectively [1,8]. However, DCE retains a role in the Tz for risk stratification of ambiguous lesions (PI-RADS 3) and, possibly, for the prediction of CSPca (Gleason equal to or more than 7) towards the IPca. Both statements were retrospectively evaluated in a database of n = 286 lesions (N = 213 patients) with a histological ground truth based on an MRI-guided prostate biopsy (S1 Fig).

In the PI-RADS™v2, DCE is binomially evaluated as "positive" or "negative," based on the fast-arterial wash-in phase, and can influence the T2w score only in case of ambiguous (PI-RADS 3) lesions in the Tz. We analyzed the predictive value of DCE in triaging ambiguous lesions as well as in predicting CSPca in PI-RADS scores 4 and 5 (abbreviated as "all PI-RADS scores"). All steps in data analysis were performed for (i) all cancers (IPca+CSPca), and (ii) CSPca while respecting the individualities of the Pz and Tz (Table 1). The number of correct predictions (TP and TN) amongst the malignant lesions was high, and the DCE sensitivity for Pz/Tz was 82/77% for all cancers, 88/78% for CSPca (Table 1). However, DCE was associated with a high number of false positives (i.e., benign lesions classified as cancers), which, especially in the Tz, outnumbered the TP predictions (Table 1).

Although CSPca is usually highly vascularized, with a vivid, ultrafast kinetic in the early wash-in phase of the DCE (Fig 1A.i, 1A.ii, 1B and 1C), high-grade cancers can be associated with slow perfusion dynamic as well (Fig 1A.iv, 1D). Similarly, prostatitis and benign prostate hyperplasia often simulate Pca due to hypervascularization (Fig 1F.i, 1F.ii, 1G and 1H). Hence, a slow wash-in DCE such as observed in a prostatitis example (Fig 1 F.iv) does not exclude

**Table 1. Predictive value of the dynamic contrast enhancement in PI-RADSv2.** Evaluation of the DCE for (a) the prediction of CSPca in all PI-RADS scores and (b) for triaging ambiguous lesions (PI-RADS 3). DCE, Dynamic Contrast Enhancement; FN, false negative; FP, false positive; NPV, Negative Predictive Value; PI-RADS, Prostate Image Reporting and Data System; PPV, Positive Predictive Value; Se, sensitivity; Spe, specificity; TN, true negative; TP, true positive.

| Lesion classification according to DCE | All cancers | | | | Clinically relevant cancers | | | |
|---|---|---|---|---|---|---|---|---|
| | All PI-RADS scores | | PI-RADS 3 | | All PI-RADS scores | | PI-RADS 3 | |
| | PZ | TZ | PZ | TZ | PZ | TZ | PZ | TZ |
| TP | 53 | 44 | 0 | 0 | 42 | 32 | 0 | 0 |
| TN | 34 | 75 | 1 | 8 | 45 | 87 | 1 | 8 |
| FP | 17 | 38 | 4 | 6 | 23 | 42 | 4 | 6 |
| FN | 12 | 13 | 0 | 0 | 6 | 9 | 0 | 0 |
| Se | 0.82 | 0.77 | NaN | NaN | 0.88 | 0.78 | NaN | NaN |
| Spe | 0.33 | 0.34 | 0.80 | 0.43 | 0.34 | 0.33 | 0.80 | 0.43 |
| PPV | 0.61 | 0.37 | 0.00 | 0.00 | 0.48 | 0.27 | 0.00 | 0.00 |
| NPV | 0.59 | 0.75 | 1.00 | 1.00 | 0.79 | 0.82 | 1.00 | 1.00 |

cancer, and might even harbor a high-grade Gleason 7b CSPca (Fig 1A.iv). An intermediate wash-in kinetic can be a feature of both prostatitis (Fig 1F.i and 1F.ii) and a high-grade, Gleason 9 CSPca (Fig 1A.iii). This significant variance in DCE behavior explains the low DCE specificity, which accounts for Pz/Tz = 33/34% for all cancers and Pz/Tz = 34/33% for CSPca (Table 1). The small n of ambiguous (PI-RADS 3) lesions does not allow for a safe statistical result–however, all ambiguous lesions (Pz n = 5 and Tz n = 14) were benign, and in the vast majority falsely overcalled by the DCE (Table 1). All in all, DCE has a moderate-to-high sensitivity, especially for the detection of significant cancers, but a very low specificity between 33–34%, which considerably restricts the diagnostic value as a Pca biomarker.

## Dynamic contrast enhancement does not improve the accuracy of bi-parametric MRI for the first detection of prostate cancer

Due to the low DCE sensitivity, adjunct costs [32], and side effects associated with i.v. gadolinium enhancers, an enhancer-free, bi-parametric (T2w and DWI) MRI protocol was recently suggested as an alternative in prostate imaging. We analyzed the PI-RADS ROC curves with and without DCE in both Pz and Tz for the detection of Pca and CSPca, separately.

The ROC-Area Under Curve (AUC) of the T2w and DWI PI-RADS was 0.76/0.70 in the Pz and 0.75/0.75 in Tz for the merged cancer group and 0.69/0.73 (Pz), 0.67/0.69 (Tz) for CSPca, thus describing a moderate diagnostic accuracy (Fig 2A, 2E for all cancers, Fig 2C and 2G for CSPca). Equally, the ROC-AUC of the quantitative ADC was in Pz/Tz 0.73/0.70 for all cancers and 0.76/0.63 for CSPca, respectively, showing no significant differences from DWI-PI-RADS (Fig 2A, 2B, 2E and 2F).

PI-RADS™v2 is determined by the DWI score for the Pz and the T2w score for the Tz; however, our analysis reveals no statistically significant differences between both sequences, $p > 0.05$, chi-squared test for both the Pz and Tz regardless of the malignancy level (Tables 2 and 3). Hence, non-enhanced MR-sequences performed equivocally for the diagnosis of IPca and CSPca in our database.

Next, we evaluated the standalone diagnostic accuracy of the DCE, T2w, and DWI/ADC for the peripheral and transitional zone. The DCE ROC-AUC for the Pz for all cancers/CSPca was 0.63/0.69, significantly lower compared to the T2w for the detection of all cancers, $p = 0.020$, chi-squared test. In the separate CSPca analysis, the DCE standalone performance was equivalent to T2w and DWI, $p > 0.05$, chi-squared test (Fig 2A, Table 2). In the Tz, the DCE-AUC was 0.61/0.59 for all cancers/CSPca, also considerably lower compared to T2w and

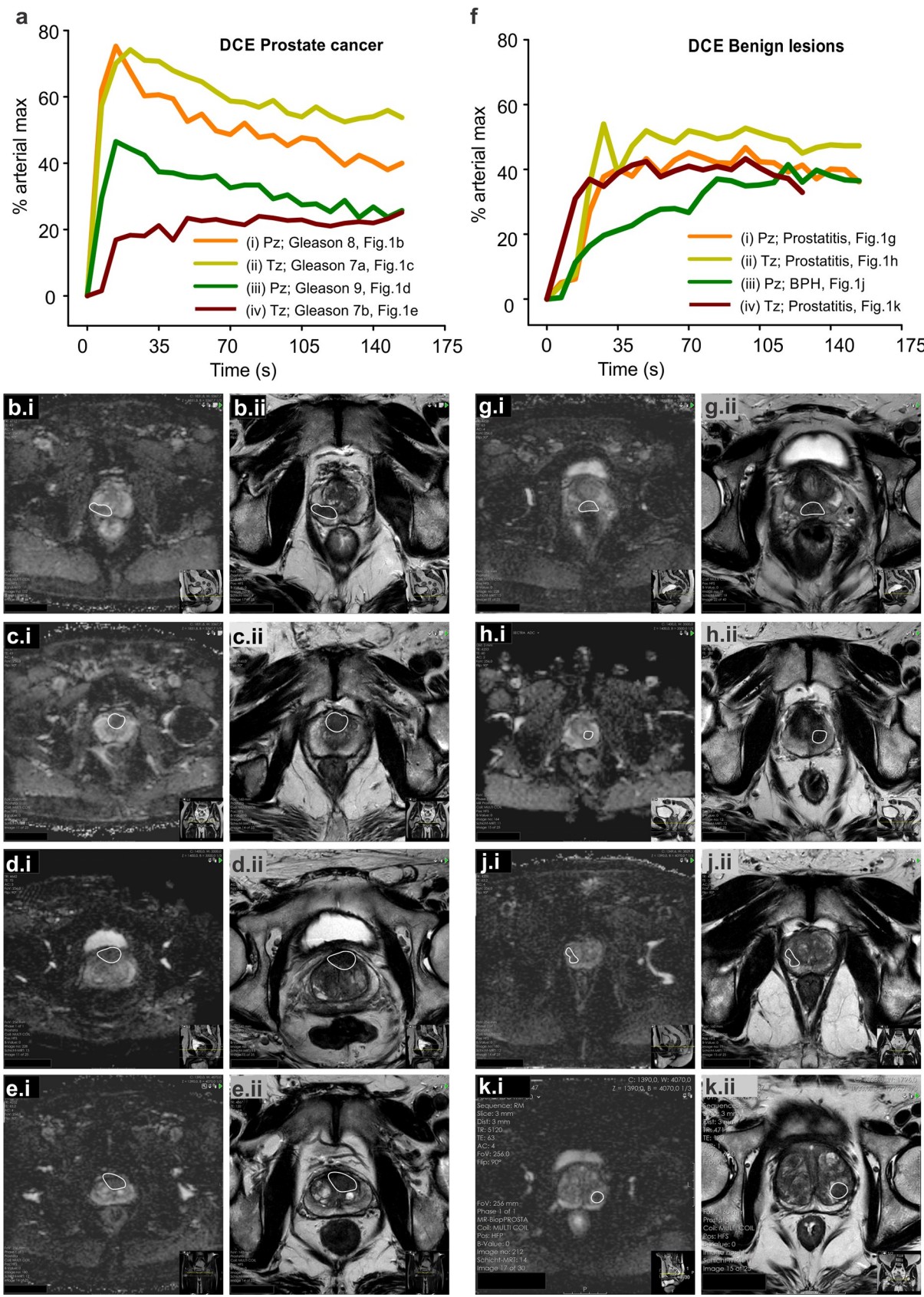

**Fig 1. Sample images of prostate cancer and benign lesions with dynamic contrast enhancement.** (A) Dynamic Contrast Enhancement (DCE) from sample patients with clinically significant prostate cancer (CSPca). (A.i, B.i, B.ii) DCE, Apparent Diffusion Coefficient (ADC) and T2w, respectively, of a peripheral zone (Pz), Gleason 8 CSPca, (A.ii, C.i, C.ii) DCE, ADC and T2w of a transitional zone (Tz), Gleason 7a CSPca, (A.iii, D.i, D.ii) DCE, ADC and T2w of a Pz, Gleason 9 CSPca and (A.iv, E.i, E.ii) DCE, ADC and T2w of a Tz, Gleason 7b CSPca. Fast wash-in and wash-out after 15 seconds, as shown in (A.i) and (A.ii), is characteristic for CSPca. However, even high-grade tumors might appear with a slow DCE, such as in (A.iv). (F) Dynamic Contrast Enhancement (DCE) from sample patients with benign prostate lesions. (F.i, G.i, G.ii) DCE, ADC and T2w of a prostatitis in Pz, respectively (F.ii, H.i, H.ii) DCE, ADC and T2w of a prostatitis in Tz (F.iii, J.i, J.ii) DCE, ADC and T2w of a benign prostate hyperplasia in Pz and (F.iv, K.i, K.ii) DCE, ADC and T2w of a prostatitis with atypical small acinar proliferation (ASAP) in Tz.

DWI for the detection of all cancers, $p = 0.001$, chi-squared test. In the separate CSPca analysis, the DCE standalone performance was equivalent to T2w and DWI, $p > 0.05$, chi-squared test (Fig 2E, Table 3). Hence, the standalone accuracy of DCE was inferior to the T2w and DWI for the diagnosis of all prostate cancers and equal to T2w/DWI for CSPca in the Pz and the Tz.

Analysis of variants between benign lesions, IPca (Gleason 3+3) and CSPca (Gleason equal or more than 7) prostate cancers (S2 Fig) showed that, although ADC is significantly reduced in both IPca and CSPca compared to benign prostate pathologies ($p < 0.05$, Kruskal Wallis ANOVA on ranks with Dunn´s post hoc test), it was still not specific enough to differentiate

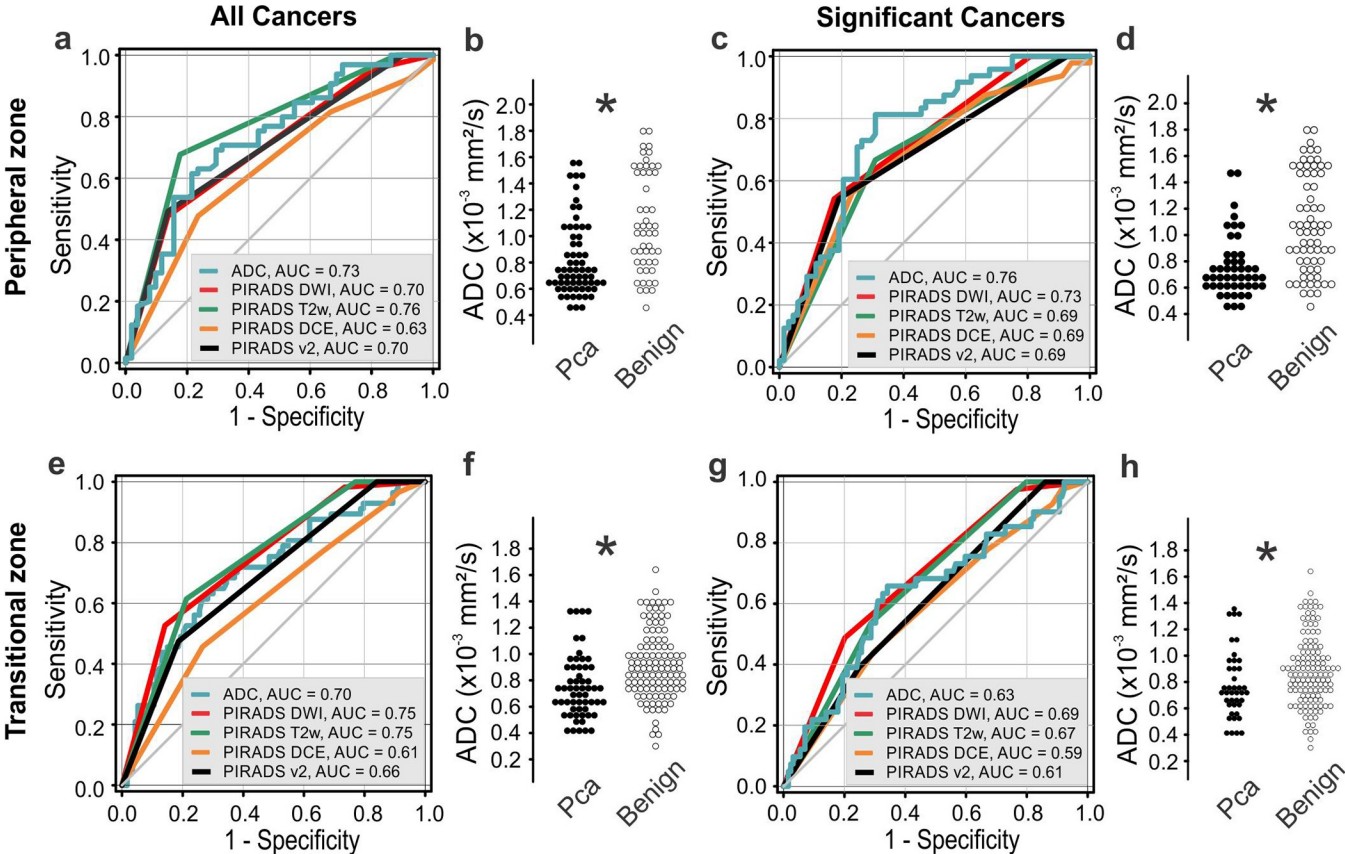

**Fig 2. Diagnostic accuracy for different mpMRI contrasts.** Receiver Operating Characteristics (ROC) for the PI-RADS DWI (red), T2w (green) and DCE (orange) compared to the quantitative ADC (blue) (A) ROC for the peripheral zone (Pz), IPca and CSPca merged as "all cancers", (B) ADC for all cancers in the Pz, $p < 0.05$, Mann Whitney rank-sum test (C) ROC for the CSPca in the Pz, (D) ADC for CSPca in the Pz, $p < 0.05$, Mann Whitney rank-sum test, (E) ROC for all cancers in the transitional zone (Tz) (F) ADC for all cancers in the Tz, $p < 0.05$, Mann Whitney rank-sum test, (G) ROC for CSPca in the Tz and (H) ADC for CSPca in the Tz, $p < 0.05$, Mann Whitney rank-sum test. There was no statistically significant difference between the diagnostic accuracy of the mpMRI parameters for the detection of both IPca and CSPca in the peripheral and transitional zone, $p > 0.05$ pairwise chi-squared test. ADC, Apparent Diffusion Coefficient; CSPca, clinically significant prostate cancer; DCE, Dynamic Contrast Enhancement; DWI, Diffusion-Weighted Imaging; IPca, insignificant prostate cancer; PI-RADS, Prostate Image Reporting and Data System; T2w, T2 weighted imaging.

**Table 2. Receiver operating characteristics for mpMRI contrasts for the detection of (i) all and (ii) clinically significant prostate cancers in the peripheral zone.**

| PI-RADS | N(m/b) | AUC ± SE | p | Cut-off | Se (%) | 95% CI | Spe (%) | 95% CI |
|---|---|---|---|---|---|---|---|---|
| DWI_all | 116(65/51) | 0.70 ± 0.04 | = 0.0003 | 4 | 95 | 87–99 | 19 | 0–33 |
| | | | | 5 | 47 | 35–60 | 86 | 74–94 |
| DWI_CSPca | 116(48/68) | 0.73 ± 0.04 | < 0.0001 | 4 | 100 | 93–100 | 19 | 11–30 |
| | | | | 5 | 54 | 39–69 | 82 | 71–91 |
| T2w_all | 116(65/51) | 0.76 ± 0.04 | < 0.0001 | 3 | 100 | 95–100 | 4 | 0–14 |
| | | | | 4 | 100 | 95–100 | 12 | 4–24 |
| | | | | 5 | 68 | 55–79 | 82 | 69–92 |
| T2w_ CSPca | 116(48/68) | 0.69 ± 0.04 | 0.0004 | 3 | 100 | 93–100 | 3 | 0–10 |
| | | | | 4 | 100 | 93–100 | 9 | 3–18 |
| | | | | 5 | 67 | 52–80 | 69 | 57–80 |
| DCE_all | 116(65/51) | 0.63 ± 0.05 | 0.016 | 3 | 92 | 83–98 | 7 | 0–19 |
| | | | | 4 | 82 | 70–90 | 33 | 21–48 |
| | | | | 5 | 48 | 35–61 | 77 | 63–87 |
| DCE_CSPca | 116(48/68) | 0.69 ± 0.05 | 0.0007 | 3 | 94 | 83–99 | 9 | 3–18 |
| | | | | 4 | 88 | 75–95 | 34 | 22–46 |
| | | | | 5 | 56 | 41–71 | 75 | 65–86 |
| PI-RADS with DCE_all | 116(65/51) | 0.70 ± 0.04 | 0.0002 | 4 | 100 | 95–100 | 10 | 3–21 |
| | | | | 5 | 49 | 37–62 | 86 | 74–94 |
| PI-RADS with DCE_CSPca | 116(48/68) | 0.69 ± 0.04 | 0.0004 | 4 | 100 | 93–100 | 7 | 2–16 |
| | | | | 5 | 54 | 39–69 | 81 | 70–90 |
| ADC_all | 116(65/51) | 0.73 ± 0.05 | < 0.0001 | 857 ($10^{-6}$ mm²/s) | 69 | 57–80 | 71 | 56–83 |
| ADC_CSPca | 116(48/68) | 0.76 ± 0.04 | < 0.0001 | 857 ($10^{-6}$ mm²/s) | 83 | 67–91 | 69 | 57–80 |

PI-RADS without (w/o) DCE is determined by the DWI score in the peripheral zone (Pz). Pairwise ROC-curve testing with chi-square test for all cancers: DWI vs. T2w $p$ = 0.152; DWI vs. DCE $p$ = 0.223; T2w vs. DCE $p$ = 0.023; PI-RADS with vs. PI-RADS w/o DCE $p$ = 0.959. Pairwise ROC-curve testing with chi-square test for CSPca: DWI vs. T2w $p$ = 0.494; DWI vs. DCE $p$ = 0.435; T2w vs. DCE $p$ = 0.885, PI-RADS with vs. PI-RADS w/o DCE $p$ = 0.095. ADC, Apparent Diffusion Coefficient; DCE, Dynamic Contrast Enhancement; DWI, Diffusion-Weighted Images; DWI/T2/DCE/PI-RADSv2/ADC_all, diagnostic accuracy for all cancers; DWI/T2/DCE/PI-RADSv2/ADC_CSPca, diagnostic accuracy for clinically significant cancers; m/b, malignant/benign; mpMRI, multiparametric Magnetic Resonance Imaging; PCa, Prostate Cancer; PI-RADS, Prostate Image Reporting and Data System; T2w, T2 weighted images.

between high- and low-grade cancers ($p$>0.05). Previous literature suggested that the DCE could bridge the diagnostic gap of ADC and facilitate the differentiation between low- and high-grade cancers [24–26]. By selecting the clinically significant cancers (Fig 2C, 2D, 2G and 2H) we could observe that the DCE ROC-AUC was not superior to the DWI/ADC either in the Pz (Fig 2C, 2D, and Table 2) or Tz (Fig 2G, 2H and Table 3), $p$>0.05, chi-squared test. We conclude that DCE had no significant added diagnostic value to T2w or DWI/ADC for the detection of CSPca.

To directly answer the question of whether DCE has an added value to mpMRI, we tested head-to-head the diagnostic accuracy of PI-RADSv2 with and without DCE (Fig 3) for all cancers and CSPca in the Pz and Tz, respectively. The AUC of PI-RADS with DCE was 0.70/0.69/0.66/0.61 for the Pz (all cancers) /Pz (CSPca) / Tz (all cancers) / Tz (CSPca), respectively. Omitting the DCE did not statistically influence the diagnostic accuracy of PI-RADS (AUC = 0.70/0.73/0.75/0.67), $p$ = 0.96/0.09/0.08/0.14, chi-squared test respectively (Fig 3A, 3B, 3E and 3F). We further questioned whether the DCE could be beneficial for the stratification of small tumors. Hence, each Pz and Tz database (Fig 3I and 3J as a histogram) was split into two subgroups setting 11 mm as a threshold for the smaller tumors. The ROC analysis (Fig 3C, 3D, 3G and 3H) showed that DCE did not improve the PIRADS performance, and, in the case

**Table 3. Receiver operating characteristics for mpMRI contrasts for the detection of (i) all and (ii) clinically significant prostate cancers in the transitional prostate zone.**

| PI-RADS | N(m/b) | AUC ± SE | p | Cut-off | Se (%) | 95% CI | Spe (%) | 95% CI |
|---|---|---|---|---|---|---|---|---|
| DWI_all | 170(57/113) | 0.75 ± 0.03 | <0.0001 | 3 | 100 | 94–100 | 1 | 0–4 |
| | | | | 4 | 98 | 91–100 | 27 | 19–36 |
| | | | | 5 | 53 | 39–66 | 86 | 78–92 |
| DWI_CSPca | 170(41/129) | 0.69 ± 0.04 | 0.0002 | 3 | 100 | 91–100 | <1 | 0–4 |
| | | | | 4 | 98 | 87–100 | 24 | 16–32 |
| | | | | 5 | 49 | 33–65 | 80 | 72–86 |
| T2w_all | 170(57/113) | 0.75 ± 0.04 | <0.0001 | 3 | 100 | 91–100 | 2 | 1–6 |
| | | | | 4 | 100 | 94–100 | 23 | 16–32 |
| | | | | 5 | 61 | 48–74 | 79 | 70–86 |
| T2w_CSPca | 170(41/129) | 0.67 ± 0.04 | 0.001 | 3 | 100 | 91–100 | 2 | 0–5 |
| | | | | 4 | 100 | 91–100 | 20 | 14–28 |
| | | | | 5 | 54 | 37–69 | 71 | 63–79 |
| DCE_all | 170(57/113) | 0.61 ± 0.04 | 0.044 | 3 | 93 | 83–98 | 12 | 7–20 |
| | | | | 4 | 77 | 64–87 | 34 | 25–43 |
| | | | | 5 | 46 | 32–59 | 74 | 64–81 |
| DCE_CSPca | 170(41/129) | 0.59 ± 0.05 | 0.087 | 3 | 93 | 80–98 | 12 | 7–18 |
| | | | | 4 | 78 | 62–89 | 33 | 25–41 |
| | | | | 5 | 44 | 29–60 | 71 | 62–78 |
| PI-RADS with DCE_all | 170(57/113) | 0.66 ± 0.04 | 0.036 | 3 | 100 | 94–100 | 4 | 0–9 |
| | | | | 4 | 100 | 94–100 | 16 | 10–19 |
| | | | | 5 | 47 | 34–61 | 81 | 73–88 |
| PI-RADS with DCE_CSPca | 170(41/129) | 0.61 ± 0.04 | 0.029 | 3 | 100 | 91–100 | 32 | 1–8 |
| | | | | 4 | 100 | 91–100 | 14 | 8–21 |
| | | | | 5 | 39 | 24–56 | 75 | 67–82 |
| ADC_all | 170(57/113) | 0.70 ± 0.04 | <0.0001 | 758 ($10^{-6}$ mm$^2$/s) | 63 | 49–76 | 72 | 62–80 |
| ADC_CSPca | 170(41/129) | 0.63 ± 0.05 | 0.01 | 772 ($10^{-6}$ mm$^2$/s) | 66 | 49–80 | 66 | 57–74 |

PI-RADS without (w/o) DCE is determined by the T2w score in the transitional zone (Tz). Pairwise ROC-curve testing with chi-square test for all cancers: DWI vs. T2w $p$ = 0.951; DWI vs. DCE $p$ = 0.001; T2w vs. DCE $p$ = 0.0001; PI-RADS with vs. PI-RADS w/o DCE $p$ = 0.0084. Pairwise ROC-curve testing with chi-square test for CSPca: DWI vs. T2w $p$ = 0.654; DWI vs. DCE $p$ = 0.092; T2w vs. DCE $p$ = 0.117, PI-RADS with vs. PI-RADS w/o DCE $p$ = 0.143. ADC, Apparent Diffusion Coefficient; DCE, Dynamic Contrast Enhancement; DWI, Diffusion-Weighted Images; DWI/T2/DCE/PI-RADSv2/ADC_all, diagnostic accuracy for all cancers; DWI/T2/DCE/PI-RADSv2/ADC_CSPca, diagnostic accuracy for clinically significant cancers; m/b, malignant/benign; mpMRI, multiparametric Magnetic Resonance Imaging; PCa, Prostate Cancer; PI-RADS, Prostate Image Reporting and Data System; T2w, T2 weighted images.

of small Tz tumors, even significantly worsened the bi-parametric prognostic value, $p$ = 0.04 chi-squared test (Fig 3G). For the small and large lesions in Pz (Fig 3C and 3D) and for the large Tz lesions (Fig 3H) the effect of DCE was equivocal ($p$ = 0.54/0.20/0.71) Thus, DCE did not increase the diagnostic accuracy of PI-RADS for IPca or CSPca in our database.

## The character of lesions overcalled by the dynamic contrast enhancement

The tissue perfusion, as assessed by the DCE, reflects the degree of neovascularization and vessel permeability [33]. Since neoangiogenesis cascades can be induced by cancer, benign prostate hyperplasia, and chronic inflammation through different mechanisms [34–36], we questioned whether DCE tends to overcall particular types of benign lesions. The retrospective analysis showed that approximately 65–70% of all benign lesions and ca. 80% of the Atypical Small Acinar Proliferation (ASAP) and Gleason 3+3 lesions were overcalled by the DCE,

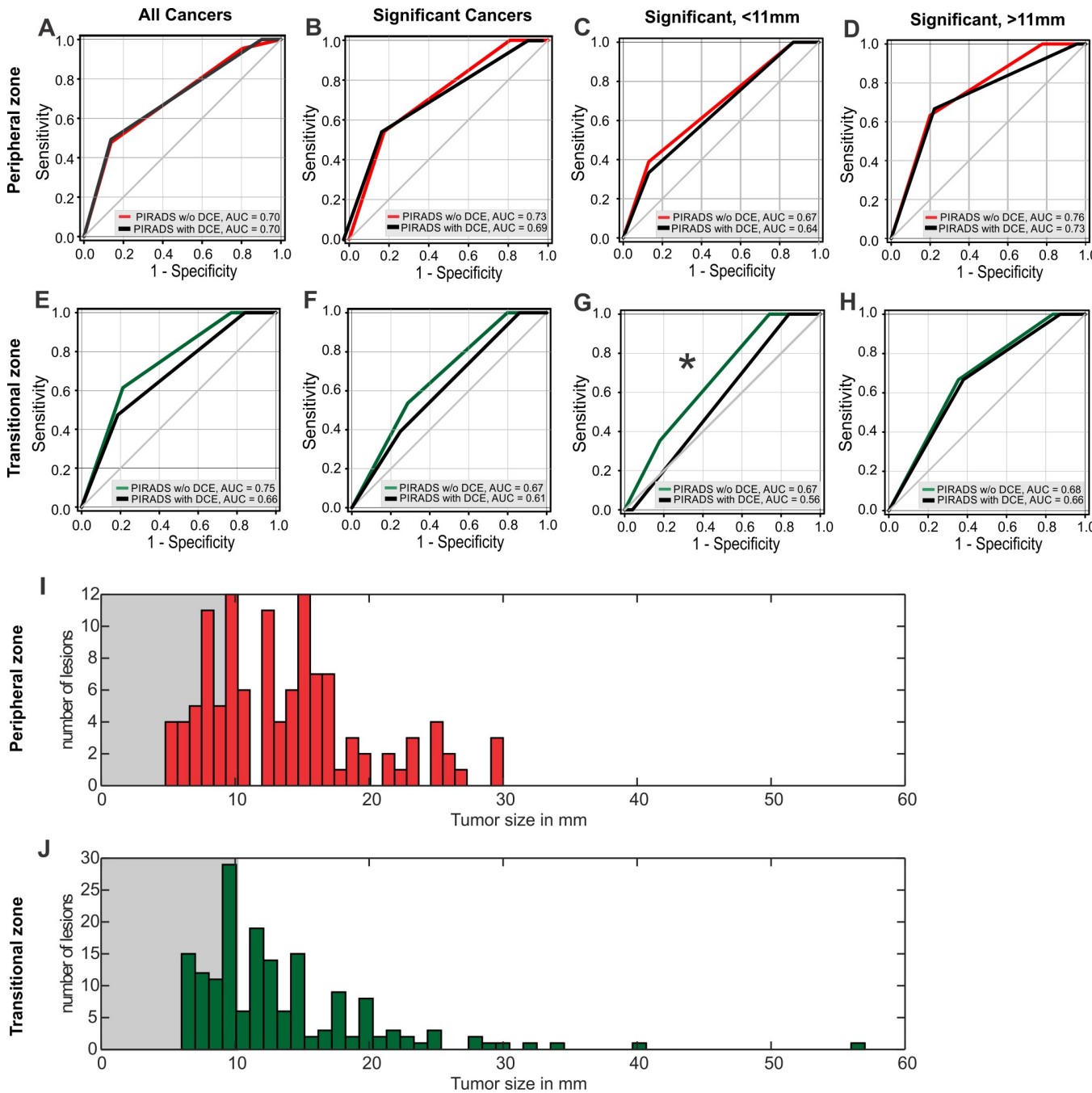

**Fig 3. Diagnostic accuracy of PI-RADS with and without DCE.** Receiver Operating Characteristic (ROC) for PI-RADS with DCE (black) and without DCE (red or green). (A) ROC for the peripheral zone (Pz) IPca and CSPca merged as "all cancers,", $p = 0.96$, AUC CI 95% 0.62–0.78 for both PIRADS with and without DCE. (B) ROC for CSPca in the Pz, $p = 0.09$, AUC CI 95% 0.65–0.81 PIRADS without DCE and 0.61–0.78.(C) ROC for CSPca sized < 11 mm in the Pz, $p = 0.54$, AUC CI 95% 0.54–0.80 without and 0.51–0.78 with DCE. (D) ROC for CSPca sized > 11 mm in the Pz, $p = 0.20$, AUC CI 95% 0.66–0.85 without and 0.63–0.83 with DCE. (E) ROC for all cancers in the transitional zone (Tz), $p = 0.08$, AUC CI 95% 0.68–0.81 without and 0.62–0.76 with DCE, (F) ROC for CSPca in the Tz, $p = 0.14$, AUC CI 95% 0.59–0.75 without and 0.53–0.69 with DCE, (G) ROC for CSPca sized < 11 mm in the Tz, $p = 0.04$, AUC CI 95% 0.56–0.79 without and 0.50–0.62 with DCE, (H) ROC for CSPca sized > 11 mm in the Tz, $p = 0.71$, AUC CI95% 0.58–0.78 without and 0.56–0.77 with DCE.All curve differences were tested with a pairwise chi-squared test. (I) Histogram of the lesion size in the Pz. The size is expressed as the maximum diameter in paraxial T2w sections. (J) Histogram of the lesion size in Tz. ADC, Apparent Diffusion Coefficient; CSPca, clinically significant prostate cancer; DCE, Dynamic Contrast Enhancement; DWI, Diffusion-Weighted Imaging; IPca, insignificant prostate cancer; PI-RADS, Prostate Image Reporting and Data System; T2w, T2 weighted imaging.

showing no preference in the Pz or Tz (S3 Table). All in all, our study failed to associate a specific benign prostate pathology with the DCE-overcalling.

## Discussion

This study concludes that the DCE had no added value in the diagnostic accuracy of PI-RADS; hence, selected candidates could be screened for prostate cancer with a faster, gadolinium-free protocol. This trend [14,15] is supported by recently published, retrospective databases, such as the research of Scherrer et al. with N = 344, [37], Junker et al. with N = 236 [21], Cristel et al. [24] with N = 313 and De Visschere [17] encompassing N = 257 patients. Various methodological differences can be spotted between our study and the reports mentioned above, such as the use of superficial vs. endorectal coil, MRI-guided biopsy vs. MRI/TRUS biopsy [37], whole-mount preparation as the gold standard [17] and variation of the DWI b-values [21,24]. Technical differences between studies impede, on the one hand, the head-to-head comparison; on the other hand, they reveal the reproducibility of the main result under different conditions. Various other groups with smaller databases also converge towards the opinion that the DCE is not necessary for prostate mpMRI [16,18–20]. The metanalysis of Woo et al. [20] and Alabousi et al. [23] concluded that "the performance of bpMRI was similar to that of mpMRI in the (first) diagnosis of prostate cancer."

Despite the cumulating evidence, DCE is a subject of debate and a current guideline recommendation in prostate imaging [2,38]. The hypothesis that DCE could facilitate the differentiation between low- and high-grade cancers [24–26] was not confirmed by our study, as we did not find any significant advantage of the DCE ROC-AUC towards the T2w and DWI for both IPca and CSPca. Numerous recent reports proactively support the role of the DCE as problem-solver for inexperienced readers. Gatti et al. (N = 68) suggest that PIRADS without DCE is a valid alternative for expert readers, whereas less experienced ones need DCE to improve the sensitivity [28]. This hypothesis could not be tested in our study because we had access only to the final conjoint report of the experienced and inexperienced reader. Alternative DCE-validation approaches might improve diagnostic accuracy. Sun et al. [39] mention that DCE performed better than T2w and DWI in volumetric Pca studies, and Parra et al. analyze the DCE image entropy to classify prostate cancer based on the behavior of "DCE-microdomains" [25]. Altogether, the DCE remains a highly debated field in prostate mpMRI and a persisting challenge for the PI-RADS steering committee.

One of the main findings of our study is that DCE has a moderate-to-good sensitivity but a very low specificity for prostate cancer, especially for the CSPca. The low sensitivity of DCE was already commented on by earlier studies, such as by Kozlowski et al. [40], which concludes that DCE reduces the specificity of T2w for a small gain of sensitivity. DCE is an indirect index of vascular permeability, which is a feature of neovessels occurring in Pca but also in benign prostate hyperplasia (BPH), atypical hyperplasia, and chronic inflammation [35]. Neoangiogenesis is a putative link between inflammation and cancer [34], albeit activated through different mechanisms in each situation [36]. Hence, overlapping neovessel formation in benign and malignant conditions could explain the low specificity of DCE. Besides, neovascularization might not necessarily be a feature of Pca [41–43]. Vessel markers such as the Vascular Endothelial Growth Factor Receptor 2 are not coherently elevated in Pca patients [44,45]. A genome meta-analysis has failed to correlate Pca with any VEGF polymorphism, whereas such association was proven for bladder cancer [46]. Hence, neovascularization might not necessarily be a feature of the CSPca, which could partially explain the low DCE specificity [33].

PIRADSv1 was a powerful albeit complex scoring system, leading to an interrater agreement rate of 0.39–0.64 [47–49]. Simplification of the scoring system in PIRADSv2 significantly

improved the interrater agreement: from 0.64 to 0.70 in the study of Becker et al. [47], from 0.39 to 0.56 in the study of Tewes et al. [48] and from 1.33 (Bland-Altman statistics for sum-score) to 0.41 in Krishna et al. [49]. Interobserver agreement studies for mp- vs. bi-parametric MRI were not performed yet, a further simplification is nevertheless quite likely to improve the interrater agreement. Krishna et al. [49] show that the kappa rate of DWI for the peripheral zone (equaling PIRADSv2 without DCE) was 0.51, improved compared to the PIRADSv2 kappa = 0.41. This highlights the necessity for future dedicated study designs towards an unbiased, rater-experience-weighted [28] evaluation of the interrater performance between mp- and bi-parametric MRI.

Within the disadvantages of this study, as well as of the majority of similar cited studies, is the retrospective character, which can provide only a low level of evidence, even if performed as a multicenter study [18] or meta-analysis [20,23]. Our study includes a low number of PI-RADS 3 lesions, especially in the Tz. Nevertheless, equally low proportions of PIRADS 3 lesions were observed in other studies (De Visschere et al. [17], 8%). Moreover, databases with a higher percentage of ambiguous lesions (Cristel et al. [24], 17%; Junker et al. [21] 20%) come up with a low DCE specificity.

Even though in the meanwhile numerous studies converge to the conclusion that gadolinium could be omitted without hampering the diagnostic accuracy of MRI, the use of gadolinium enhancer is a matter of debate [27] and a recommendation in the current prostate imaging guidelines [7,8,50]. With our contribution, we opt to strengthen the cumulating evidence towards the optimization of the upcoming guidelines for prostate diagnostics.

## Supporting information

**S1 Table. Baseline characteristics of patients and lesions.** N/n = patients/lesions. All values are presented as mean/standard deviation. PI-RADS 2 lesions were higher classified and biopsied according to the PI-RADSv1 system, then downgraded to PI-RADS 2 upon PI-RADSv2 re-evaluation. CSPca; Clinically Significant Prostate cancer, IPca; Insignificant Prostate cancer, mpMRI; multiparametric MRI.
(DOCX)

**S2 Table. Prostate MRI technical parameters.** T2 TSE, T2-weighted Turbo Spin Echo; DWI, Diffusion-Weighted Imaging; DCE, Dynamic Contrast Enhancement; TE, Echo Time; TR, Repetition Time; FH, Foot-Head direction; RL, Right-Left direction; AP, Anterior-Posterior direction.
(DOCX)

**S3 Table. The character of lesions overcalled by the dynamic contrast enhancement.**
(DOCX)

**S1 Fig. Participant flow and study design.** *N* stands for patient and *n* for lesion number. Accuracy of PI-RADS with and without Dynamic Contrast Enhancement (DCE). Clinically significant prostate cancers (CSPca) were analyzed for the peripheral (Pz) and transitional (Tz) prostate zones separately. DCE, Dynamic Contrast Enhancement; mpMRI, multiparametric Magnetic Resonance Imaging; PI-RADS, Prostate Image Reporting and Data System; T2w, T2-weighted images; DWI, Diffusion-Weighted Images; PCa, Prostate Cancer.
(TIF)

**S2 Fig. Apparent diffusion coefficient of benign and malignant prostate lesions.** N = 168/ 33/85 for benign, IPca, and CSPca lesions. Peripheral and transitional zones merged. $P < 0.05$, Kruskal-Wallis ANOVA on ranks. ADC, Apparent Diffusion Coefficient; IPca, insignificant

prostate cancer; CSPca, clinically significant prostate cancer.
(TIF)

**S1 File. Original dataset in .xlsx spreadsheet.** Sheet 1 "Raw Data". Data arranged in columns. (A) patient index, (B) distance between mpMRI and biopsy in days, (C) patient age in years rounded up to the closest integer (D) prostate zone encoding, **p** for peripheral and **z** for transitional lesions, (E) MRI field strength in T, (F) lesion size as the maximal diameter in paraaxial sections, in mm, (G) Apparent Diffusion Coefficient in $mm^2/s \times 10^{-6}$. (H) PIRADS T2-w score (I) PIRADS DWI score, (J) PIRADS DCE score (according to PIRADS version 1) (K) PIRADS v1 total score (L) PIRADS v2 total score. (M) number of biopsy probes (N) Gleason score (O) Gleason a or b (P) Histology index: 1 = prostate cancer, PCa; 2 = prostatitis; 3 = benign prostate hyperplasia; 4 = prostate tissue without pathology; 6 = periprostatic tissue; 7 = Atypical Small Acinar Proliferation. Sheet 2 "PIRADS v2 DCE". Data arranged in columns. Dynamic Contrast Enhancement (DCE) score according to PIRADS v2 criteria, 1 = negative and 2 = positive. (A) patient index for peripheral zone lesions (B) DCE PIRADSv2 score for peripheral zone lesions, (C) patient index for transitional zone lesions (B) DCE PIRADSv2 score for transitional zone lesions.
(XLSX)

## Author Contributions

**Conceptualization:** Ilinca Cosma, Ansgar Malich, Ismini Papageorgiou.

**Data curation:** Sven Winzler, Ismini Papageorgiou.

**Formal analysis:** Ilinca Cosma, Cornelia Tennstedt-Schenk, Ismini Papageorgiou.

**Investigation:** Ilinca Cosma, Cornelia Tennstedt-Schenk, Ismini Papageorgiou.

**Methodology:** Ilinca Cosma, Marios Nikos Psychogios, Alexander Pfeil, Ulf Teichgraeber, Ansgar Malich.

**Project administration:** Ansgar Malich, Ismini Papageorgiou.

**Resources:** Cornelia Tennstedt-Schenk, Ulf Teichgraeber, Ansgar Malich.

**Software:** Marios Nikos Psychogios, Alexander Pfeil, Ulf Teichgraeber, Ansgar Malich.

**Supervision:** Ansgar Malich, Ismini Papageorgiou.

**Validation:** Ansgar Malich, Ismini Papageorgiou.

**Visualization:** Ilinca Cosma, Sven Winzler.

**Writing – original draft:** Ilinca Cosma.

**Writing – review & editing:** Alexander Pfeil, Ansgar Malich, Ismini Papageorgiou.

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
