## [Decision Letter · Decision Letter 0]

3 Dec 2019

PONE-D-19-30220

The role of gadolinium in magnetic resonance imaging for early prostate cancer diagnosis: a diagnostic accuracy study

PLOS ONE

Dear Dr. Papageorgiou,

Thank you for submitting your manuscript to PLOS ONE. After careful consideration, we feel that it has merit but does not fully meet PLOS ONE’s publication criteria as it currently stands. Therefore, we invite you to submit a revised version of the manuscript that addresses the points raised during the review process.

We would appreciate receiving your revised manuscript by Jan 17 2020 11:59PM. To enhance the reproducibility of your results, we recommend that if applicable you deposit your laboratory protocols in protocols.io, where a protocol can be assigned its own identifier (DOI) such that it can be cited independently in the future. For instructions see: http://journals.plos.org/plosone/s/submission-guidelines#loc-laboratory-protocols

We look forward to receiving your revised manuscript.

Kind regards,

Pascal A. T. Baltzer, M.D.

Academic Editor

PLOS ONE

Journal Requirements:

2. We note that you have reported significance probabilities of 0 in places. Since p=0 is not strictly possible, please correct this to a more appropriate limit, eg 'p<0.0001'.

Reviewers' comments:

Reviewer's Responses to Questions

**Comments to the Author**

1. Is the manuscript technically sound, and do the data support the conclusions?

Reviewer #1: Yes

2. Has the statistical analysis been performed appropriately and rigorously? 

Reviewer #1: Yes

3. Have the authors made all data underlying the findings in their manuscript fully available?

Reviewer #1: Yes

4. Is the manuscript presented in an intelligible fashion and written in standard English?

Reviewer #1: Yes

5. Review Comments to the Author

Reviewer #1: Congratulations to this overall well written manuscript, which confirms the results of several prior studies in the field.

Some minor points:

- MR-guided biopsy: Was an additional systematic biopsy performed (TRUS, transperineal?).

- Additionally to the Gleason score, is it possible for the authors provide data on tumor volume estimates? (either by MR segmentation or biopsy core length)

- Readers: Please provide initials of the two readers (or names if not one of the authors) and years since board certification (or years in training)

- Please provide exact p-values instead of dichotomized >/< 0.05 in the results section.

- Consider omitting "Another critical point is the lack of healthy subjects since only patients with a level of suspicion were screened and biopsied." I fail to see how this is a limitation.

- Another point that might add to the discussion is the simplification of DCE from PIRADS v1 to v2 and studies showing increased interreader agreement. Might biparametric MRI further improve interreader agreement?

- Consider adding CIs to Fig 3. Otherwise the figures are very nice and appealing.

- I applaud the authors for providing the data on a granular, per-lesion level. Although in the interest of a short review turnover time, I have not verified the results myself, this will certainly help interested readers to do so, and other researchers when accruing data for meta analyses.

6. PLOS authors have the option to publish the peer review history of their article (what does this mean?). If published, this will include your full peer review and any attached files.

Reviewer #1: No

---

## [Author Response · Author response to Decision Letter 0]

8 Dec 2019

Response to the reviewer(s)

Reviewer #1: Congratulations to this overall well-written manuscript, which confirms the results of several prior studies in the field.

Re: We sincerely thank the reviewer for the time spent on our manuscript and for the relevant comments that improve the manuscript´s readability. We were glad to cover the requested issues in the following point-by-point response. 

001-MR-guided biopsy: Was an additional systematic biopsy performed (TRUS, transperineal?).

Re-001: Indeed, most of the patients included in this study were referred to our department upon persisting suspicion for malignancy after a negative ambulant TRUS-biopsy. Rare exceptions count for less than 10% of the available dataset. This selection line reflects on the disproportionally large number of Tz and small lesions, which are less likely to allocate without targeting. After an MRI-guided prostate biopsy, no further TRUS was performed. In N= 6 cases, however, the MRI-guided biopsy was repeated within a time interval of 612±231 (mean, std) days months because of a persisting elevation of the PSA. The treatment decision was based on the second biopsy. 

The methods’ part was updated accordingly (L120-127). 

002- Additionally to the Gleason score, is it possible for the authors to provide data on tumor volume estimates? (either by MR segmentation or biopsy core length)

Re-002: We thank the reviewer for this relevant comment and the opportunity to discuss how the tumor volume might be connected to the DCE-influence of the PIRADS predictive value. We upgraded Figure 3 and added histograms of the tumor size for each prostate zone separately (Fig. 3 I, J). The lesion size was measured as the maximal diameter in paraxial T2w sections. 

Moreover, the reviewer´s comment motivated us to analyze the impact of DCE for different tumor sizes in the Pz and Tz, setting 11 mm as an arbitrary limit for “small” and “large” lesions (Figure 3 C, D, G, H). Interestingly, the addition of DCE significantly worsens the PIRADS prognostic value for small Tz tumors. In the other groups, the effect remains equivocal. The main text and Figure 3 were updated respectively (L319-L326 and Fig. 3 files and caption). 

003- Readers: Please provide initials of the two readers (or names if not one of the authors) and years since board certification (or years in training)

Re-003: The initials and experience of the readers were added in text, L143-144. 

004- Please provide exact p-values instead of dichotomized >/< 0.05 in the results section.

Re-004: The exact p-values were provided the results section, L319 and L326

005- Consider omitting "Another critical point is the lack of healthy subjects since only patients with a level of suspicion were screened and biopsied." I fail to see how this is a limitation.

Re-005: We thank the reviewer for this comment. The relevant text was removed (L414-415)

006- Another point that might add to the discussion is the simplification of DCE from PIRADS v1 to v2 and studies showing increased interreader agreement. Might biparametric MRI further improve interreader agreement?

Re-006: The reviewer raises a crucial point for the clinical implementation of PIRADS – the interrater agreement. PIRADSv1 was a powerful albeit complex scoring system, leading to an interrater agreement rate of 0.39 – 0.64 (Becker et al., 2017; Tewes et al., 2016; Krishna et al., 2017). Simplification of the scoring system in PIRADSv2 significantly improved the interrater agreement: from 0.64 to 0.70 in the study of Becker et al., from 0.39 to 0.56 in the study of Tewes et al. and from 1.33 (Bland-Altman statistics for sum-score) to 0.41 in Krishna et al. Interobserver agreement studies for mp- vs. bpMRI were not performed yet, a further simplification is nevertheless quite likely to improve the interrater agreement. Krishna et al. show that the kappa rate of DWI for the peripheral zone (equaling PIRADSv2 without DCE) was 0.51, improved compared to the PIRADSv2 kappa = 0.41. This highlights the necessity for dedicated study designs for an unbiased, experience-weighted (Gatti et al. 2019) evaluation of the interrater performance between mp- and bpMRI. 

The discussion was upgraded accordingly (L 401-411) 

007- Consider adding CIs to Fig 3. Otherwise, the figures are very nice and appealing.

Re-007: CI at 95% were added in the legend of Figure 3, L328-343

008- I applaud the authors for providing the data on a granular, per-lesion level. Although in the interest of a short review turnover time, I have not verified the results myself, this will certainly help interested readers to do so, and other researchers when accruing data for meta-analyses.

Re-008: We are pleased to read the reviewer´s opinion, that the supplementation of the original dataset is a sine-qua-non for qualitative scientific communication. We are grateful to the editor for supplying a data-exchange platform. We will be glad to support any further implementation of our dataset in a meta-analysis or other systematic review format.

---

## [Editor Report · Decision Letter 1]

12 Dec 2019

The role of gadolinium in magnetic resonance imaging for early prostate cancer diagnosis: a diagnostic accuracy study

PONE-D-19-30220R1

Dear Dr. Papageorgiou,

Dear colleagues!

We are pleased to inform you that your manuscript has been judged scientifically suitable for publication and will be formally accepted for publication once it complies with all outstanding technical requirements.

With kind regards from Vienna,

Pascal A. T. Baltzer, M.D.

Academic Editor

PLOS ONE
---

## [Editor Report · Acceptance letter]

16 Dec 2019

PONE-D-19-30220R1 

The role of gadolinium in magnetic resonance imaging for early prostate cancer diagnosis: a diagnostic accuracy study 

Dear Dr. Papageorgiou:

I am pleased to inform you that your manuscript has been deemed suitable for publication in PLOS ONE. Congratulations! Your manuscript is now with our production department. 

With kind regards,

on behalf of

Dr. Pascal A. T. Baltzer 

Academic Editor

PLOS ONE